# Fractal Dimension, Circularity, and Solidity of Cell Clusters in Liquid-Based Endometrial Cytology Are Potentially Useful for Endometrial Cancer Detection and Prognosis Prediction

**DOI:** 10.3390/cancers16132469

**Published:** 2024-07-06

**Authors:** Toshimichi Onuma, Akiko Shinagawa, Tetsuji Kurokawa, Makoto Orisaka, Yoshio Yoshida

**Affiliations:** 1Department of Obstetrics and Gynecology, Faculty of Medical Sciences, University of Fukui, Fukui 910-1193, Japan; sngw@u-fukui.ac.jp (A.S.); orisaka@u-fukui.ac.jp (M.O.); yyoshida@u-fukui.ac.jp (Y.Y.); 2Department of Obstetrics and Gynecology, Fukui-ken Saiseikai Hospital, Fukui 918-8503, Japan; kurotetu@u-fukui.ac.jp

**Keywords:** endometrial neoplasms, cytodiagnosis, cytological techniques, image processing, computer assisted, fractals, prognosis

## Abstract

**Simple Summary:**

Various diagnostic methods exist for EC, with traditional biopsy being common but invasive. Endometrial cytology using liquid specimens shows promise for non-invasive detection. Morphological features like fractal dimension, circularity, and solidity in cell clusters could potentially enhance diagnostic accuracy and predict prognosis. Circularity and fractal dimension demonstrated significant associations with EC and AEH, regardless of age and cytology results. An ROC analysis revealed improved diagnostic accuracy when combining fractal dimension with cytology, particularly in menopausal age groups. Moreover, circularity, solidity, and fractal dimension may serve as prognostic indicators for endometrial cancer. This study showed that the morphological characteristics of cell clusters in liquid-based cytology may improve endometrial cancer diagnosis and predict prognosis.

**Abstract:**

Endometrial cancer (EC) in women is increasing globally, necessitating improved diagnostic methods and prognosis prediction. While endometrial histology is the conventional approach, liquid-based endometrial cytology may benefit from novel analytical techniques for cell clusters. A clinical study was conducted at the University of Fukui Hospital from 2012 to 2018, involving 210 patients with endometrial cytology. The liquid-based cytology images were analyzed using cell cluster analysis with Image J software. Logistic regression, ROC analysis, and survival analysis were employed to assess the diagnostic accuracy and prognosis between cell cluster analysis and EC/atypical endometrial hyperplasia (AEH). Circularity and fractal dimension demonstrated significant associations with EC and AEH, regardless of age and cytology results. The ROC analysis revealed improved diagnostic accuracy when combining fractal dimension with cytology, particularly in menopausal age groups. Lower circularity and solidity were independently associated with poor overall survival, while higher fractal dimension values correlated with poorer overall survival in Grades 2 and 3 endometrial cancers. The combination of circularity and fractal dimension with cytology improved diagnostic accuracy for both EC and AEH. Moreover, circularity, solidity, and fractal dimension may serve as prognostic indicators for endometrial cancer, contributing to the development of more refined screening and diagnostic strategies.

## 1. Introduction

A growing number of women worldwide have been diagnosed with endometrial cancer (EC) [1,2], more notably in Japan [3]. Approximately 5–10% of patients present with irregular genital bleeding post-menopause [4]. Various techniques have been developed for the early diagnosis of EC from tissue, cell, serum, and urine specimens [2]. A biopsy of the endometrium is currently the most commonly used method [5]. Traditionally, dilatation and curettage (D&C) was the primary means of evaluating the endometrium [6]. Hysteroscopic biopsy was associated with the highest endometrial cancer grade agreement compared to D&C and Pipelle [7]. On the other hand, it has been reported that D&C is more consistent with postoperative pathology results than Pipelle or hysteroscopy, independent of BMI [8]. Therefore, it remains unclear whether D&C or hysteroscopy is superior in diagnosing endometrial cancer. D&C is an invasive procedure [9]. While aspiration tissue biopsy using Pipelle is an excellent method for the diagnosis of EC [10], and is effective for lesions that have spread throughout the uterus, its effectiveness for focal lesions is unknown [6]. The diagnostic accuracy of Pipelle endometrial sampling is associated with the amount of endometrial tissue surface, with a minimal cut-off value of 35 mm^2^ required to classify an endometrial sample as conclusive [11]. However, there is no standard for ensuring that the aspiration tissue biopsy is properly collected. Only patients with an external uterine opening large enough to allow the insertion of this collection device can be examined. Furthermore, endometrial biopsy using Pipelle had problems, with approximately 15% of patients reporting pain and 34% reporting anxiety during the procedure [12].

Endometrial cytology is a noninvasive method for detecting EC [9]. However, the test is not often used owing to its unreliability, with previous studies reporting a sensitivity of 78% for the detection of EC [13]. Endometrial cytology using the liquid specimen method improves specimen quality compared to the conventional direct smear method, as observed in cervical cancer screening [14]. In fact, it is common practice in cervical cytology to use liquid specimens rather than conventional dry specimens [15,16,17]. The usefulness of this method in endometrial cytology has been confirmed by a number of studies, demonstrating its superior diagnostic accuracy compared to conventional methods [18,19,20,21,22]. Its results are comparable to those of an endometrial biopsy [22].

The Yokohama System (TYS) is a diagnostic criterion for endometrial cytology using the liquid specimen method. It has a detection sensitivity of 96.7% for EC and atypical endometrial hyperplasia (AEH) [21]. Furthermore, in a similar study comparing endometrial cytology using TYS and aspiration biopsy, the sensitivity for detection of EC and AEH was 92.2% and 85.2%, respectively. The specificity was 98.9% and 98.5%, respectively [22]. There has been a significant improvement in the quality of endometrial cytology using the liquid specimen method. In Japan, endometrial cytology is widely used as a screening test for uterine endometrial cancer [21].

The architecture of cell clusters is an important factor in endometrial cytology. In cervical cytology, cell morphology is classified using the Bethesda system according to the nuclear atypia and nuclear/cytoplasmic ratio [23,24,25]. In the first step of TYS, cell cluster atypia is used to diagnose [21,26]. The cell clusters with irregular protrusion patterns and nuclear overlapping of three or more layers at low microscopic magnification are important findings that are directly suggestive of endometrial carcinoma [21,26].

In recent years, fractal dimension has been utilized in many different areas. It represents the complexity of the data in relation to the spatial scale; the higher the fractal dimension, the more complex the data [27]. Fractal dimension is a measure that quantifies the complexity of biological structures and phenomena [27]. It has been applied to molecular biological analysis and pathology for tumor, vascular, and neural lesions [27]. A previous study has shown that the fractal dimension from the surface of endometrial adenocarcinoma tissue varies with the histological grade [28]. Nevertheless, the use of fractal dimension in liquid endometrial cytology has not been investigated.

Circularity and solidity are indicators of structural irregularities [29]. Nuclear circularity and solidity are useful in the identification of high-grade urothelial carcinoma in urine cytology [29]. Similarly, distinguishing between spindle cell melanoma and desmoplastic melanoma is critical because of the differences in metastatic rate and prognosis. The quantification of nuclear circularity can be used for this purpose [30]. However, the use of circularity and solidity in liquid endometrial cytology is yet to be explored.

The storage of a variable amount of well-preserved cells in liquid-based cytology allows for the application of immunocytochemical and molecular techniques [31]. These advanced methods may improve the evaluation of treatment response, resistance to targeted therapies, and prognosis prediction in thyroid, lung, and breast carcinoma, as well as malignant effusions [31]. While novel, such analyses require additional costs.

The morphology of cancer is associated with its prognosis. In breast cancer tissue, the number, circularity, and total perimeter of tumor nests are independent prognostic factors for the patients’ 5-year disease-free survival [32]. In colorectal cancer tissue, the complexity ratio calculated from colorectal cancer nodule perimeter and area is associated with significantly worse disease-free survival [33]. Molecular biological prognostic markers have been reported for EC [34]. The updated 2023 staging of endometrial cancer includes various molecular classifications that reflect the prognosis of the endometrial cancer [35,36]. However, prognostic prediction using morphologic features of cell clusters in liquid-based endometrial cytology has not been established.

Therefore, the purpose of this study was to determine whether the circularity, solidity, and fractal dimension of the cell clusters in liquid-based cytology are related to EC or AEH and can be applied to enhance the accuracy of cytology diagnosis. Further, we aimed to determine whether this method could predict the prognosis of EC.

## 2. Materials and Methods

### 2.1. Study Population

This study was conducted in adherence to the tenets of the Declaration of Helsinki. Patients whose histological diagnosis was performed after endometrial cytology at the University of Fukui Hospital (Eiheiji, Yoshida District, Fukui, Japan) from 2012 to 2018 were included in the study. For the diagnostic system of endometrial cytology, a three-tier reporting system consisting of negative, suspicious-positive, and positive results was used [37,38]. Of the patients with positive or suspicious-positive endometrial cytology, those whose histological results were normal endometrium (Normal), endometrial hyperplasia without atypia (EH), AEH, endometrioid carcinoma (EM) grade (G)1, EM G2, EM G3, and serous and clear cell carcinoma (Others) were included in the analysis. As a result, 91 endometrial cytology-suspicion-positive cases and 83 endometrial cytology-positive cases matching these criteria were included in the analysis. In this study, 26 negative cytology cases were randomly selected, resulting in an almost equal number of AEH plus EC and Normal plus EH cases for histology results. All endometrial cytology examinations were performed by a skilled cytologist and pathologist. All histological diagnoses were performed by two skilled pathologists. This study was approved by the Research Ethics Committee of University of Fukui (No. 20150015). A consent form was not obtained for the public disclosure of study information (opt-out method) [39].

### 2.2. Liquid-Based Endometrial Cytology

Endometrial cells were collected directly from the uterine cavity using an Honest Super Brush (Honest Medical Corporation, Tokyo, Japan). The brush was dipped into a vial of BD SurePath preservative fluid (BD Diagnostics, Franklin Lakes, NJ, USA). The cells were washed off the brush by rotating it [40], and automated liquid specimen processing was performed using the BD PrepMate System (BD Diagnostics, Franklin Lakes, NJ, USA), according to the manufacturer’s protocol. The cells were transferred to the BD PrepStain Density Reagent (BD Diagnostics, Franklin Lakes, NJ, USA), gravity sedimented, and subjected to Papanicolaou staining [37].

### 2.3. Obtaining and Processing Liquid-Based Endometrial Cytology Image

An optical microscope BX50 (Olympus, Tokyo, Japan) and an optical microscope camera DP20 (Olympus, Tokyo, Japan) were used to acquire endomembrane cell images. Five images were captured of the top, bottom, left, right, and center of the circular LBC slide area using a 4× objective lens. Each of the imaging positions was pre-set to avoid overlap. ImageJ (ver 1.53a, National Institute of Health, Bethesda, MD, USA) was used to analyze the captured images. The images were converted to 8-bit and then to binary (Figure 1A,B). Measurements for cell clusters were examined using the “Analyze Particle” function and the box-counting method for fractal dimensions in ImageJ. The analyzed parameters included perimeter, fit eclipse major, fit eclipse minor, angle, circularity, solidity, maximum Feret diameter, and minimum Feret diameter. The average of the five fields was used for the analysis.

### 2.4. Statistical Analyses

A one-way analysis of variance with Dunnett’s test was used to compare the continuous variables. Age was divided into two groups at ≥55 yr, considering postmenopausal patients [41]. Logistic regression analysis was performed for cell cluster analysis results adjusted by age (≥55 yr) and cytology diagnosis (positive or suspicious positive/negative) to compare EC (EM G1, EM G2, EM G3, and Others) and AEH with non-EC (Normal and EH). Receiver operating characteristic (ROC) analysis was performed to evaluate the sensitivity, specificity, and area under the curve (AUC) and determine the cutoff value. The AUCs of ROC analysis for EC and AEH diagnosis based on cytology alone were compared with those for EC + AEH diagnosis based on cytology and cell cluster analysis results. Survival curves were plotted using the Kaplan–Meier estimator, and *p* values were calculated using the log-rank test. The X-tile software (version 3.6.1) was used to determine cut-off values for categorizing high or low circularity, solidity, and fractal dimension [42]. Hazard ratios and 95% confidence intervals for circularity, solidity, and fractal dimension were evaluated using Cox proportional hazards models. Statistical analyses were performed with EZR version 1.42 (Saitama Medical Center, Jichi Medical University, Saitama, Japan), a graphical user interface for R version 4.0.0 (The R Foundation for Statistical Computing, Vienna, Austria) [43]. *p* < 0.05 was considered statistically significant.

## 3. Results

### 3.1. Cytology and Histology Diagnosis

Table 1 presents the patient characteristics, including cytology, age, and histology diagnosis. No EC/AEH was observed in the negative cytology cases. Moreover, EC cases included significantly older patients than normal cases (EM G1 *p* < 0.001, EM G2 *p* < 0.001, EM G3 *p* = 0.043, Others *p* = 0.018, respectively) (Table 1).

### 3.2. Results of the Relationship between EC and AEH Diagnosis and Cell Cluster Analysis Results

Table 2 shows the results of logistic regression analysis to determine the relationship between EC and AEH and cell cluster analysis. The mean and range values of cell cluster analysis for each histology are shown in Appendix A. Fit ellipse major, fit ellipse minor, angle, circularity, solidity, and fractal dimension were significantly associated with EC and AEH (Table 2). Cytology was divided into two groups: positive or negative, and suspicious-positive. Circularity and fractal dimension are independently associated with EC and AEH after adjusting for cytology and postmenopausal age.

### 3.3. ROC Analysis

Figure 2 shows the ROC curves for EC and AEH diagnosis through cell cluster analysis for all ages. The AUC values between 0.6 and 0.7, 0.71 and 0.8, and greater than 0.8 indicate weak predictive, satisfactory, and good predictive abilities, respectively [44]. The AUC for the diagnostic accuracy of EC and AEH is unknown due to the lack of previous studies on cell clusters. Therefore, we conducted a sample size analysis to determine if the sample size meets the following criteria: a significance level of 5%, 90% power, a two-tailed test, and a ratio of normal to abnormal cases of 1:1. Assuming an AUC of 0.7 on the ROC curve, the required number of cases was 40. Assuming an AUC of 0.65 on the ROC curve, the required number of cases was 73. Graphs depicting the relationship between thresholds and sensitivity or specificity for cell cluster measurements are provided in Appendix A. Fit ellipse major AUC 0.603 (95% CI 0.524–0.682), fit ellipse minor AUC 0.619 (95% CI 0.542–0.697), angle AUC 0.640 (95% CI 0.563–0.716), circularity AUC 0.683 (95% CI 0.609–0.756), and solidity AUC 0.651 (95% CI 0.576–0.727) had weak predictive abilities. Only fractal dimension AUC 0.722 (95% CI 0.652–0.793) had a satisfactory predictive value. The fractal dimension had a cutoff of 1.327 with a sensitivity of 82.1% and a specificity of 54.3%. Figure 3 shows the ROC curves of EC and AEH diagnosis through cell cluster analysis in postmenopausal patients. Similar to the results mentioned earlier, the fit ellipse major AUC 0.623 (95% CI 0.489–0.756), fit ellipse minor AUC 0.633 (95% CI 0.497–0.770), angle AUC 0.652 (95% CI 0.525–0.780), and solidity AUC 0.647 (95% CI 0.507–0.786) had weak predictive abilities. Moreover, the circularity AUC 0.706 (95% CI 0.582–0.829) had a satisfactory predictive ability, while the fractal dimension AUC 0.874 (95% CI 0.795–0.950) had a good predictive ability. The fractal dimension had a cutoff of 1.337 with a sensitivity of 88.2% and a specificity of 76.9%.

### 3.4. Performance Differences in Cytology Plus Cell Cluster Analysis

By comparing ROC curves, we evaluated whether adding cell cluster analysis to the cytology would increase the diagnostic accuracy of EC and AEH. Based on the results of the logistic regression and ROC analysis, this performance analysis was conducted for circularity, solidity, and fractal dimension. Solidity with cytology did not increase the AUC compared to cytology alone (Figure 4C,D). In comparison, circularity with cytology increased the AUC in the group of all ages (circularity with cytology AUC 0.840 vs. cytology AUC 0.807, *p* = 0.0426) (Figure 4A). However, this increase was not observed in the >55-year-age group with menopause (circularity with cytology AUC 0.851, cytology 0.825 *p* = 0.350) (Figure 4B). Meanwhile, fractal dimension with cytology increased the AUC compared to cytology alone in both the group with all ages (fractal dimension with cytology AUC 0.860 vs. cytology AUC 0.807, *p* = 0.002) (Figure 4E) as well as the >55-year-age group (menopause fractal dimension with cytology AUC 0.917 vs. cytology AUC 0.825, *p* = 0.002) (Figure 4F).

### 3.5. Prognostic Impact of Cell Cluster Analysis

In the cell cluster analysis, the fractal dimension had the most diagnostic impact. Therefore, we analyzed the prognostic impact of cell cluster analysis. Cutoff values were set using X-tile software [42]. Lower circularity was significantly associated with poor overall survival (OS) in all histological types (circularity cutoff 0.55, *p* = 0.002) and G1–G3 EM (circularity cutoff 0.55, *p* = 0.002) or G2 and G3 EM (circularity cutoff 0.56, *p* = 0.009) (Figure 5A–C). Similarly, lower solidity was significantly associated with poor OS in all histological types (solidity cutoff 0.58, *p* = 0.005) and G1–G3 EM (solidity cutoff 0.56, *p* = 0.002) or G2 and G3 EM (solidity cutoff 0.58, *p* = 0.005) (Figure 6A–C). No association was observed between OS and fractal dimension for any of the histologic types (fractal dimension cutoff 1.300, *p* = 0.093) or G1–G3 EM (fractal dimension cutoff 1.300, *p* = 0.166) (Figure 7A,B). However, a significant correlation between higher fractal dimension and poor OS was observed in G2 and G3 EM (fractal dimension cutoff 1.650, *p* = 0.009) (Figure 7C).

The Cox hazards regression analysis showed that circularity and solidity were associated with OS in the univariable analysis (Table 3). Circularity and solidity were independently associated with OS in multivariable analysis adjusted by age (≥55 yr), histology (endometrioid or not), and number of cell clusters.

## 4. Discussion

In this study, we investigated whether the addition of cell cluster analysis to liquid-based endometrial cytology can improve diagnostic accuracy. Logistic regression analysis showed that circularity and fractal dimension were independently associated with EC and AEH after adjustment for age and cytology diagnosis. In addition, ROC analysis of EC and AEH diagnoses in the menopausal age group showed that the fractal dimension had the highest AUC of 0.874. We further examined whether adding the results of cell cluster analysis to cytology would increase the accuracy of EC and AEH diagnoses. In the group of all ages, the AUC of circularity with cytology (0.840) was significantly higher than that of cytology alone. The AUC for the combination of fractal dimension and cytology was 0.860 for all ages and 0.917 for the menopausal age group, both significantly increased compared to the AUC of cytology alone. These results suggest that the addition of cell cluster analysis to cytology can improve diagnostic accuracy. Furthermore, higher circularity and solidity were independently associated with better OS. Thus, cell cluster analysis with liquid endometrial cytology may improve diagnostic accuracy and prognosis prediction.

Fractal analysis of cell clusters is useful for the diagnosis of EC and AEH. In this study, fit ellipse major, fit ellipse minor, angle, circularity, solidity, and fractal dimension were all significantly associated with EC and AEH. Additionally, circularity and fractal dimension demonstrated independent associations with EC adjusted for menopausal status and cytology results. Fractal analysis is useful for morphological evaluation in cytology, while quantification of nuclear circularity can be used as an adjunct diagnostic tool to distinguish between spindle cell and desmoplastic melanomas [30]. Fractal analysis using the box-counting method is useful in analyzing the chromatin complexity of cervical dysplastic cells [45]. Analysis of cell clusters in mammary puncture aspiration cytology can assist in the diagnosis of breast cancer [46]. Invasive breast cancer is composed of larger cell clusters rather than fibroadenoma. Furthermore, the fractal value of tissue fragment shape is significantly higher in invasive breast cancer than in fibroadenoma [46]. In cervical cytology, the fractal dimension of nuclei increases according to the degree of dysplasia [47]. The fractal dimensions of cervical intraepithelial neoplasia (CIN) 1, CIN 2, and CIN 3 nuclei exhibit significant differences [47]. Comparing a pathologist’s diagnosis with a prediction by fractal analysis showed a concordance rate of 87.1% [48]. Similarly, morphometric shape definitions of circularity can be used to assess nuclear membrane irregularity in high-grade urothelial carcinoma cells [29]. The results of our study suggest that circularity and fractal dimension in liquid-based cytology may be associated with EC and AEH.

Our findings indicate that adding cell cluster analysis to cytology could potentially increase the accuracy of EC and AEH diagnoses. Adding carcinoembryonic antigen staining to the cytology of malignant pleural effusions increases diagnostic sensitivity compared to that with cytology alone [49]. Circularity and fractal dimension combined with endometrial cytology may be able to detect EC cases more accurately than endometrial cytology alone. Thus, circularity and fractal dimension may contribute to the early detection of EC.

Cell cluster analysis may be useful in predicting prognosis. In our study, lower circularity and solidity were independently associated with poor OS. The circularity of tumor nests could be a useful parameter to predict the prognosis of early-stage breast invasive ductal carcinoma [32]. Lower circularity and solidity indicate that cell clusters have a more irregular protrusion pattern [29,50], indicating EC. The fractal dimension may be used to predict the prognosis for endometrial cancer. Fractal dimension measured in malignant tumors is associated with a poor prognosis [51]. Fractal dimension of chromatin structure is an independent, unfavorable prognostic factor for OS in small-cell neuroendocrine lung carcinomas [52]. Another study indicates that fractal dimensions of the nuclear chromatin were independent, unfavorable prognostic factors for survival in patients with melanoma [53]. To our knowledge, our study is the first to demonstrate that the circularity, solidity, and fractal dimension of cell clusters in endometrial cytology are associated with a poor prognosis of EC.

Endometrial cell cluster analysis at low magnification could be more useful than that at high magnification. Fractal analysis of nuclei is performed with a defined region of interest [48]. In histopathology, normal and malignant tissues may exist far from each other. It is necessary to set a region of interest to adapt the analysis. In endometrial cytology using the liquid specimen method, blood and mucus contamination is minimal, and cells are evenly distributed in a thin layer on the slide [54,55]. Therefore, this study did not select specific cell clusters for analysis. In other words, there was no selection bias. Furthermore, in the fractal analysis of breast cancer tissue, no difference between benign and malignant tumors was found in the entire image at the high magnification of 400×; however, a significant difference in fractal dimension was found at the low magnification of 40× [56,57]. Thus, observations at high magnification may be inappropriate for extracting cell cluster features.

Postmenopausal bleeding is an important factor in the diagnosis of EC [58]. In this study, the AUC of fractal analysis for EC diagnosis was high at menopausal age. In postmenopausal age, cytological diagnosis had sufficient diagnostic accuracy. On the other hand, the diagnostic accuracy was low in premenopausal age. This might be due to morphological changes in the endometrium caused by menstrual cycles. There is often morphological confusion between endometrial glandular breakdown and malignancy. Immunostaining of insulin-like growth factor II mRNA-binding protein 3 can be added to the liquid specimen method to identify the origin of the cells, which is difficult to distinguish by morphology [59]. Thus, in liquid endometrial cytology, the addition of immunostaining has the potential to improve the accuracy of cytology.

FIGO 2023 staging uses the molecular classification proposed by The Cancer Genome Atlas (TCGA), which classifies endometrial carcinomas into four categories based on the molecular classification: polymerase ε (POLE, ultramutated), microsatellite instability (MSI, hypermutated), copy number low, and copy number high [35,36]. The molecular classification is linked to the prognosis of endometrial cancer [35,36]. Multiomics approaches offer a comprehensive view of the molecular landscape of EC, aiding in the identification of markers and therapeutic targets. These methods integrate data from the genome, transcriptome, proteome, and epigenome, providing a broader understanding of the disease [60]. By incorporating a morphological approach, as demonstrated in our study, we will establish a foundation for innovative diagnostic and therapeutic methods.

The study has some limitations. Firstly, it is a single-center study, which may limit the generalizability of the findings. Conducting a multi-center study would help validate the results across diverse populations and settings. Additionally, this study did not analyze the relationship between cell clusters, which could provide valuable insights into disease mechanisms and progression. Many factors are used for diagnosis in endometrial cytology, including structural atypia of cell clusters, thickness of the cluster, background, and cell atypia [61]. The shape of a cell cluster is extremely complex and may contain many characteristics of malignancy. AI-based diagnostic techniques can discover useful features of malignancy in endometrial cytology that are not detectable by humans. Previous reports have analyzed the nuclear morphology of liquid-based endometrial cytology with an artificial neural network based on a multi-layer perceptron and shown its usefulness in the diagnosis of EC [62]. However, these are still underdeveloped areas that require further research. Image analysis techniques utilizing artificial intelligence could offer a more comprehensive understanding of cellular interactions and patterns within endometrial cytology samples.

## 5. Conclusions

In conclusion, cell cluster analysis of endometrial cytology, including parameters such as circularity, solidity, and fractal analysis, is useful for the diagnosis of EC. Specifically, fractal analysis may be added to cytology to improve diagnostic accuracy. Furthermore, circularity, solidity, and fractal dimension may predict EC prognosis. With the recent rise in EC incidence, incorporating cell cluster analysis into endometrial cytology may increase its utility.

## Figures and Tables

**Figure 1 cancers-16-02469-f001:**
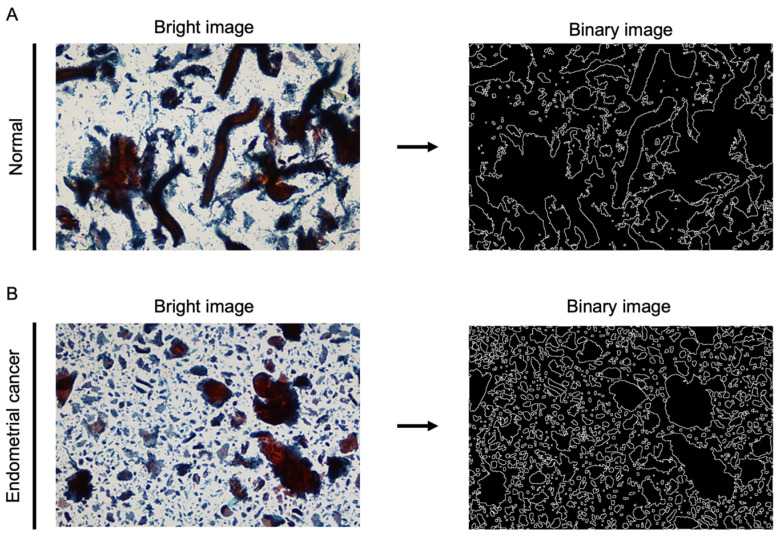
Processing of a liquid-based endometrial cytology image. (**A**,**B**) A bright image of the normal endometrium or endometrial cancer was obtained using a 4× objective lens. It was then converted to an 8-bit binary image using ImageJ software.

**Figure 2 cancers-16-02469-f002:**
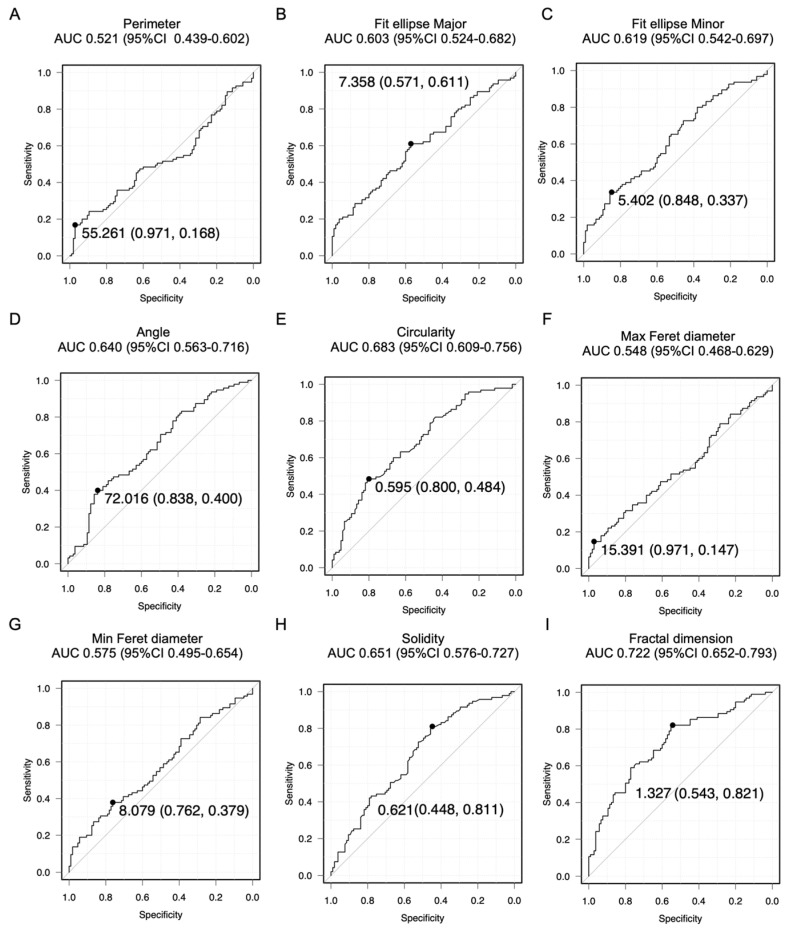
ROC curve analysis for all ages. The numbers on the graph represent following value: Cutoff (Specificity, Sensitivity). ROC curves for the detection of AEH and EC (EM G1, EM G2, EM G3, serous cancer, and clear cell cancer) based on the following parameters: (**A**) perimeter, (**B**) fit ellipse major, (**C**) fit ellipse minor, (**D**) angle, (**E**) circularity, (**F**) Max Feret diameter, (**G**) Min Feret diameter. (**H**) solidity, and (**I**) fractal dimension. Endometrioid cancer (EM); grade (G); atypical endometrial hyperplasia (AEH); receiver operating characteristic (ROC); maximum (Max); minimum (Min).

**Figure 3 cancers-16-02469-f003:**
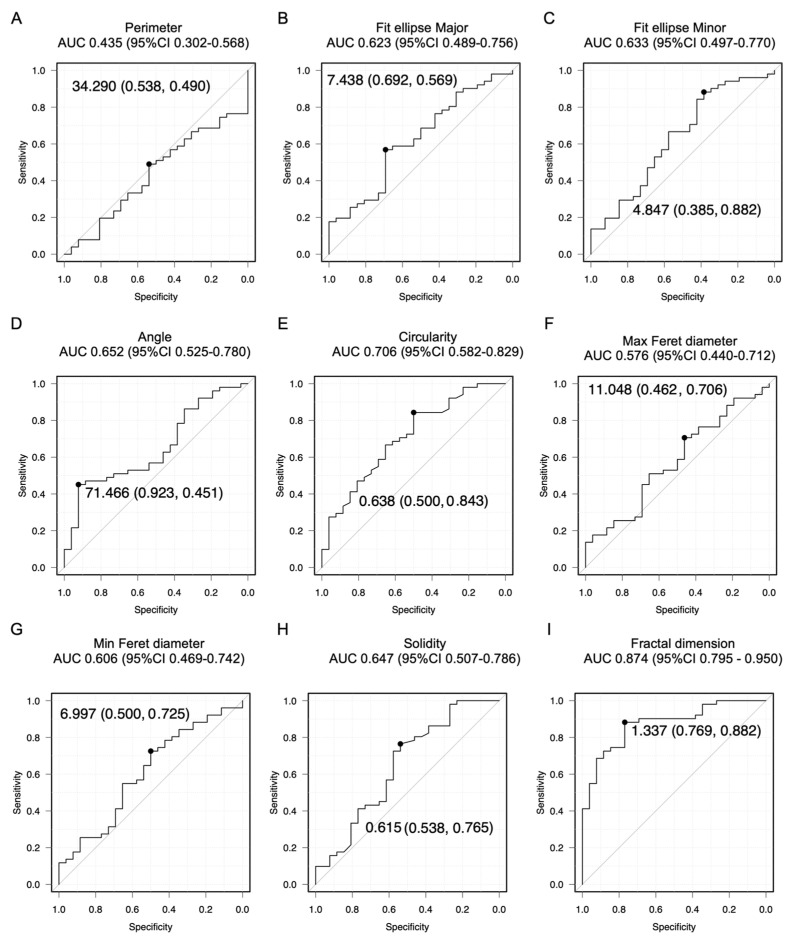
ROC curve analysis for ages over 55 yr. The numbers on the graph represents the following value: Cutoff (Specificity, Sensitivity). ROC curves for the detection of AEH and EC (EM G1, EM G2, EM G3, serous cancer, and clear cell cancer) based on the following parameters: (**A**) perimeter, (**B**) fit ellipse major, (**C**) fit ellipse minor, (**D**) angle, (**E**) circularity, (**F**) Max Feret diameter, (**G**) Min Feret diameter, (**H**) solidity, and (**I**) fractal dimension. Endometrioid cancer (EM); grade (G); atypical endometrial hyperplasia (AEH); receiver operating characteristic (ROC); maximum (Max); minimum (Min).

**Figure 4 cancers-16-02469-f004:**
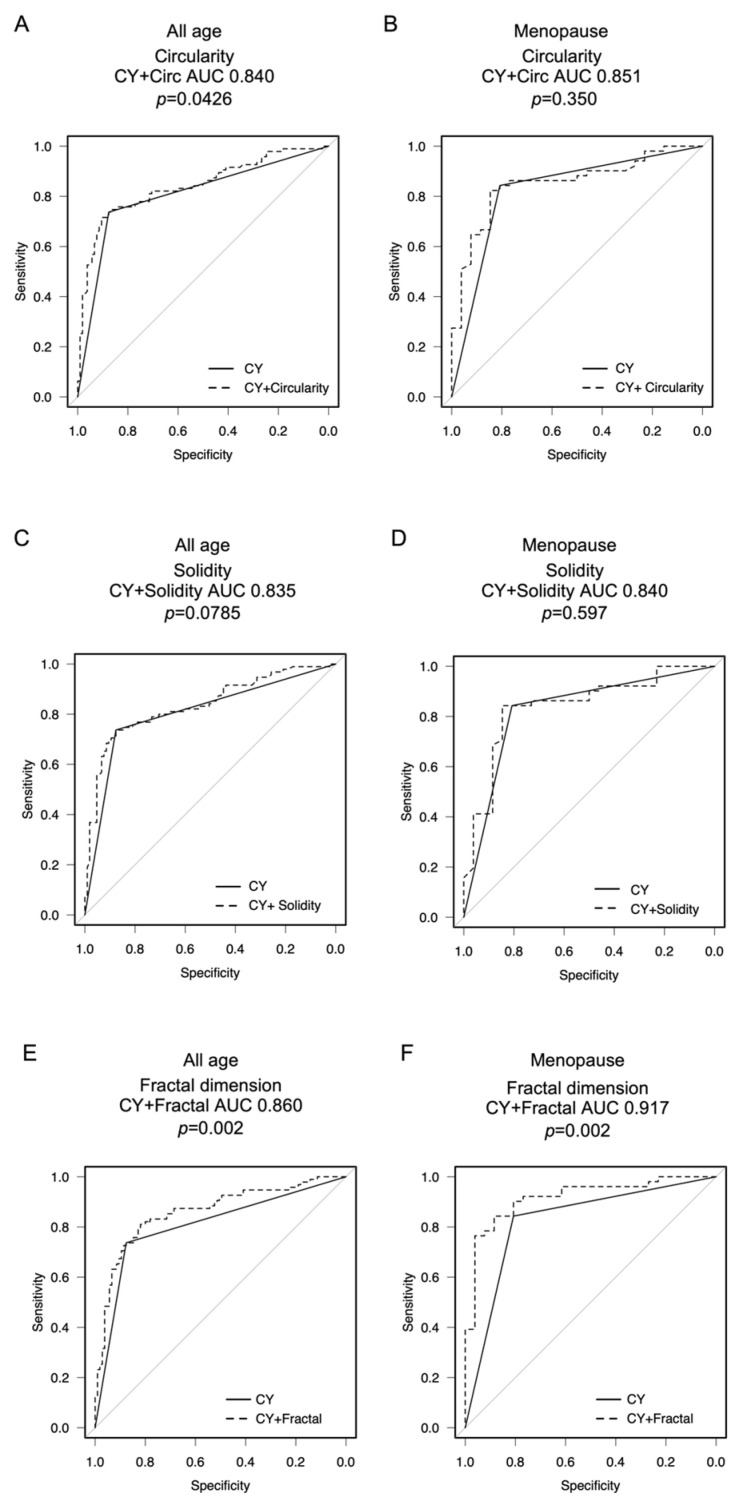
Performance analysis of cell clusters with cytology for detecting the EC and AEH. EC includes endometrioid cancer, serous cancer, and clear cell cancer. The EC includes endometrioid cancer, serous cancer, and clear cell cancer. The AUC of the ROC curve for cytology diagnosis was compared to that for cytology diagnosis plus circularity, solidity, and fractal dimension in (**A**,**C**,**E**) for all ages and (**B**,**D**,**F**) for ages > 55 yr, respectively, atypical endometrial hyperplasia (AEH), endometrial cancer (EC), and CY: cytology diagnosis.

**Figure 5 cancers-16-02469-f005:**
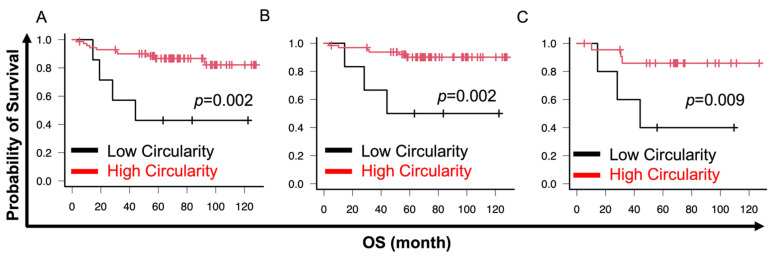
Prognostic impact of circularity. Cutoffs were set using X-tile software. EC includes EM, serous cancer, and clear cell cancer. (**A**–**C**) Lower circularity was significantly associated with poor OS in EC and EM as well as EM G2 and G3. (**A**) EC; (**B)** EM; (**C**) EM G2 and G3. Endometrial cancer (EC); endometrioid cancer (EM); grade (G).

**Figure 6 cancers-16-02469-f006:**
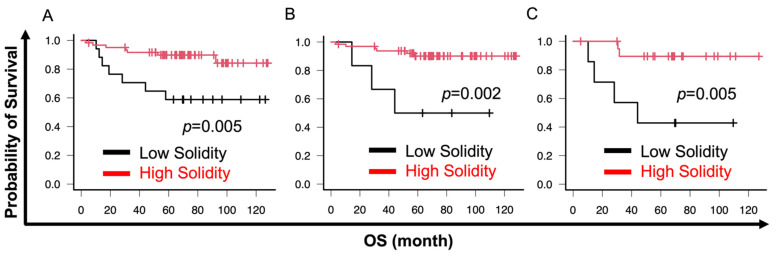
Prognostic impact of solidity. Cutoffs were set using X-tile software. EC includes EM, serous cancer, and clear cell cancer. (**A**–**C**) Lower solidity was significantly associated with poor OS in EC and EM as well as EM G2 and G3. (**A**) EC; (**B**) EM; (**C**) EM G2 and G3. Endometrial cancer (EC); endometrioid cancer (EM); grade (G).

**Figure 7 cancers-16-02469-f007:**
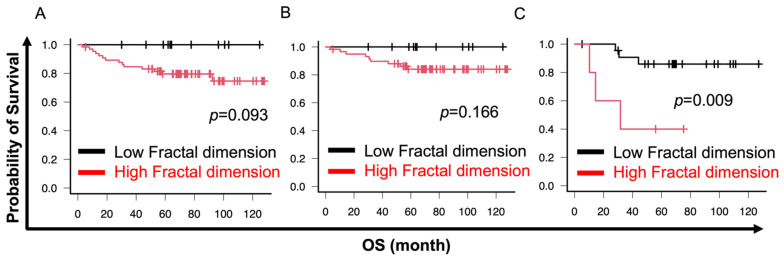
Prognostic impact of fractal dimensions. Cutoffs were set using X-tile software. EC includes EM, serous cancer, and clear cell cancer. (**A**,**B**) The results showed no association between OS and fractal dimension for EC and EM. (**C**) Higher fractal dimension was significantly associated with poor OS in EM G2 and G3. (**A**) EC; (**B**) EM; (**C**) EM G2 and G3. Endometrial cancer (EC); endometrioid cancer (EM); grade (G).

**Table 1 cancers-16-02469-t001:** Patient characteristics.

	Normal (*n* = 92)	EH (*n* = 13)	AEH (*n* = 17)	EM G1 (*n* = 43)	EM G2 (*n* = 18)	EM G3 (*n* = 10)	^†^ Others (*n* = 7)
Age in yr (SD)	49.9 (12.2)	45.9 (6.9)	46.2 (10.6)	*** 59.2 (10.6)	** 63.1 (13.8)	* 60.3 (8.0)	* 63.4 (13.6)
Cytology							
Negative	25	1	0	0	0	0	0
Suspicious Positive	57	9	11	11	3	0	0
Positive	10	3	6	32	15	10	7

Standard deviation (SD); endometrial hyperplasia without atypia (EH); atypical endometrial hyperplasia (AEH); endometrioid carcinoma (EM); grade (G) ^†^: Serous and clear cell carcinoma; ***: Normal vs. EM G1, *p* < 0.001; **: Normal vs. EM G2, *p* = 0.001; *: Normal vs. Others, *p* = 0.022.

**Table 2 cancers-16-02469-t002:** Logistic regression analysis of the relationship between EC or AEH and cell cluster analysis.

	Unadjusted	*p*-Value	Adjusted	*p*-Value
Perimeter (95% CI)	1.02 (0.993–1.04)	0.169	1.020 (0.986–1.050)	0.281
Fit ellipse major (95% CI)	1.6 (1.16–2.22)	0.005	1.36 (0.891–2.07)	0.155
Fit ellipse minor (95% CI)	3.33 (1.56–7.09)	0.002	1.84 (0.708–4.78)	0.211
Angle (95% CI)	1.12 (1.05–1.2)	0.001	1.08 (0.994–1.180)	0.068
Circularity (95% CI)	1.17 × 10^−7^ (7.23 × 10^−11^–1.89 × 10^−4^)	>0.001	0. 841 × 10^−4^ (0.112 × 10^−7^–0.629)	0.039
Max Feret diameter (95% CI)	1.12 (0.978–1.280)	0.102	1.09 (0.919–1.3)	0.102
Min Feret diameter (95% CI)	1.32 (1.02–1.7)	0.033	1.09 (0.919–1.3)	0.317
Solidity (95% CI)	9.35 × 10^−9^ (4.1 × 10^−13^–2.13 × 10^−4^)	>0.001	0.28 × 10^−4^ (1.32 × 10^−10^–5.92)	0.094
Fractal dimension (95% CI)	79.7 (14.8–429)	>0.001	31 (3.97–242)	0.001

The EC includes endometrioid cancer, serous cancer, and clear cell cancer. Confidence interval (CI); atypical endometrial hyperplasia (AEH); endometrial cancer (EC); maximum (Max); minimum (Min).

**Table 3 cancers-16-02469-t003:** Cox regression analysis for overall survival.

	Univariable		Multivariable	
	HR (95% CI)	*p*-Value	HR (95% CI)	*p*-Value
Circularity	0.198 (0.061–0.635)	0.006	0.175 (0.051–0.595)	0.005
Solidity	0.248 (0.087–0.708)	0.009	0.184 (0.062–0.548)	0.002

Multivariable analysis was adjusted by age (≥55 yr), histology (endometrioid or not), and number of cell clusters. Confidence interval (CI).

## Data Availability

The original contributions presented in the study are included in the article/Appendix A. Further inquiries can be directed to the corresponding author.

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
