# Peer review of "Fractal Dimension, Circularity, and Solidity of Cell Clusters in Liquid-Based Endometrial Cytology Are Potentially Useful for Endometrial Cancer Detection and Prognosis Prediction"

_cancers, 2024, doi:10.3390/cancers16132469_

Round 1

Reviewer 1 Report

Comments and Suggestions for Authors

I read with great interest article "Fractal dimension, circularity, and solidity of cell clusters in liquid-based endometrial cytology are potentially useful for endometrial cancer detection and prognosis prediction".

I would recommend to add more precisely about accuracy of preoperative sampling diagnosis. Reviews showed that although hysteroscopic biopsy was mainly associated with the highest tumor grade agreement, and although D&C showed the highest overall accuracy in detecting endometrial carcinoma, the data do not therefore reliably indicate which method yields the most precise results.

I would also recommend to add more about new  FIGO classification - The performance of complete molecular classification (POLEmut, MMRd, NSMP, p53abn) is encouraged in all endometrial cancers. In 2009, FIGO removed cytology as a staging criterion from the endometrial cancer staging system, in 2023 there is not mention anything about cytology. How can your study have clinical implication - please add some discussion.

I am curious why do you think AI could offer a more comprehensive understanding of cellular interactions and patterns within endometrial tissue samples?

Author Response

Dear Reviewers,

We would like to express our deepest gratitude for your invaluable assistance during the review process of our manuscript. Your thoughtful comments and insightful suggestions have significantly contributed to the improvement of our work.

Your thorough evaluation and constructive feedback have helped us refine the clarity and coherence of our research. We sincerely appreciate the time and expertise you have generously shared with us, as your input has been instrumental in strengthening the overall quality of our manuscript.

We have significantly improved the manuscript to reflect the new changes and suggestions provided as highlighted below:

Reviewer 1:

I read with great interest article "Fractal dimension, circularity, and solidity of cell clusters in liquid-based endometrial cytology are potentially useful for endometrial cancer detection and prognosis prediction".

1. I would recommend to add more precisely about accuracy of preoperative sampling diagnosis. Reviews showed that although hysteroscopic biopsy was mainly associated with the highest tumor grade agreement, and although D&C showed the highest overall accuracy in detecting endometrial carcinoma, the data do not therefore reliably indicate which method yields the most precise results.

Response:

Thank you for your excellent suggestion. We have added a description of hysteroscopic biopsy for endometrial cancer to the manuscript, as there was previously no mention of hysteroscopy. Additionally, we have included information on the diagnostic accuracy of various methods for detecting endometrial cancer. The following revisions have been made to the specified line:

Page 2, Line 7

Hysteroscopic biopsy was associated with the highest endometrial cancer grade agreement compared to D&C and Pipelle [7]. On the other hand, it has been reported that D&C is more consistent with postoperative pathology results than Pipelle or hysteroscopy, independent of BMI [8]. Therefore, it remains unclear whether D&C or hysteroscopy is superior in diagnosing endometrial cancer.

2. I would also recommend to add more about new  FIGO classification - The performance of complete molecular classification (POLEmut, MMRd, NSMP, p53abn) is encouraged in all endometrial cancers. In 2009, FIGO removed cytology as a staging criterion from the endometrial cancer staging system, in 2023 there is not mention anything about cytology. How can your study have clinical implication - please add some discussion.

Response:

Thank you for your insightful suggestions. We have incorporated the new FIGO classification, which includes the performance of complete molecular classification (POLEmut, MMRd, NSMP, p53abn) (Berek et al. 2023; Corr et al. 2022). As you pointed out, FIGO seems to have determined that cytology is not useful in the diagnosis of endometrial cancer. While cytology does not require anesthesia and can be performed immediately in an outpatient clinic, conventional cytology is not appropriate for making a diagnosis. However, endometrial cytology using the liquid specimen method improves upon the drawbacks of conventional cytology. Our study results can be added to endometrial cytology of liquid specimens to further improve diagnostic accuracy. The new molecular classification is capable of estimating prognosis, but it requires significant cost to perform. Our study showed that endometrial cytology of liquid specimens not only improves diagnostic accuracy but also allows for further prognostic estimation. The following revisions have been made to the specified lines in the manuscript:

Page 3, Line 18.

The updated 2023 staging of endometrial cancer includes various molecular classification which reflect prognosis of the endometrial cancer [35,36].

Page 15, Line 19

FIGO 2023 staging uses the molecular classification proposed by The Cancer Genome Atlas (TCGA), which classifies endometrial carcinomas into four categories based on the molecular classification: polymerase ε (POLE, ultramutated), microsatellite instability (MSI, hypermutated), copy number low, and copy number high [35,36]. The molecular classification is linked to the prognosis of endometrial cancer [35,36]. Multiomics approaches offer a comprehensive view at the molecular landscape of EC, aiding in the identification of markers and therapeutic targets. These methods integrate data from the genome, transcriptome, proteome, and epigenome, providing a broader understanding of the disease [60]. By incorporating a morphological approach, as demonstrated in our study, will establish a foundation for innovative diagnostic and therapeutic methods.

3. I am curious why do you think AI could offer a more comprehensive understanding of cellular interactions and patterns within endometrial tissue samples?

Response:

Many factors are used for diagnosis in endometrial cytology, including structural atypia of cell clusters, thickness of the clusters, background, and cell atypia. Each of these factors, however, can be used to determine malignancy by simple criteria. The shape of a cell cluster is extremely complex and may contain many characteristics of malignancy. We believe that AI-based diagnostic techniques can discover useful features of malignancy in endometrial cytology that are not detectable by humans. Previous reports have analyzed the nuclear morphology of liquid-based endometrial cytology using artificial neural networks based on multi-layer perceptrons and demonstrated their usefulness in diagnosing endometrial cancer. (Makris et al. 2017). However, these areas are still underdeveloped and require further research. The following revisions have been made to the specified lines in the manuscript:

Page 15, Line 33

Many factors are used for diagnosis in endometrial cytology, including structural atypia of cell clusters, thickness of the cluster, background, and cell atypia [61]. The shape of a cell cluster is extremely complex and may contain many characteristics of malignancy. AI-based diagnostic techniques can discover useful features of malignancy in endometrial cytology that are not detectable by humans. Previous reports have analyzed the nuclear morphology of liquid-based endometrial cytology with artificial neural network based on multi-layer perceptron and showed its usefulness in the diagnosis of EC [62]. However, these are still underdeveloped areas that require further research. The image analysis techniques utilizing artificial intelligence could offer a more comprehensive understanding of cellular interactions and patterns within endometrial cytology samples.

References:

Berek, Jonathan S., Xavier Matias-Guiu, Carien Creutzberg, Christina Fotopoulou, David Gaffney, Sean Kehoe, Kristina Lindemann, David Mutch, Nicole Concin, and Endometrial Cancer Staging Subcommittee, FIGO Women’s Cancer Committee. 2023. “FIGO Staging of Endometrial Cancer: 2023.” International Journal of Gynaecology and Obstetrics: The Official Organ of the International Federation of Gynaecology and Obstetrics 162 (2): 383–94.

Corr, Bradley, Casey Cosgrove, Daniel Spinosa, and Saketh Guntupalli. 2022. “Endometrial Cancer: Molecular Classification and Future Treatments.” BMJ Medicine 1 (1): e000152.

Makris, Georgios-Marios, Abraham Pouliakis, Charalampos Siristatidis, Niki Margari, Emmanouil Terzakis, Nikolaos Koureas, Vasilios Pergialiotis, Nikolaos Papantoniou, and Petros Karakitsos. 2017. “Image Analysis and Multi-Layer Perceptron Artificial Neural Networks for the Discrimination between Benign and Malignant Endometrial Lesions.” Diagnostic Cytopathology 45 (3): 202–11.

Reviewer 2 Report

Comments and Suggestions for Authors

The main aims of the studies performed by authors were to determine whether circularity, solidity, and fractal dimension of the cell clusters in liquid-based cytology are related to endometrial cancer (EC) or atypical endometrial hyperplasia (AEH), and can be used to enhance the accuracy of cytology diagnosis. 

The study's second aim was to determine whether this method can predict the prognosis of endometrial carcinoma. The paper is well written, the experiments performed are logical, and the results and conclussion are well presented

The results presented by the authors in their manuscript suggest that cell cluster analysis with liquid-endometrial cytology may improve diagnostic accuracy and prognosis prediction. 

 Circularity and fractal analysis combined with endometrial cytology may contribute to detecting EC cases more accurately than endometrial cytology alone. Thus, circularity and fractal analysis may contribute to the early detection of EC and more than this, the fractal dimension has the potential to predict the prognosis for endometrial cancer.  All these represent the novelty of this article, and for these reasons, I recommend the publication of the manuscript.

One minor observation: a sentence cannot begin with an abbreviation, and in the manuscript, I detected more sentences in this situation: one on page 5 (a sentence which begins with  ''X-tile software...''), one on page 6 (a sentence which begins with: ''AUC values...'')  and one at page 21 ( sentence with begin with ''COx...'') I recommend authors to read with attention the manuscript and rewrite sentences which begin with an abbreviation.

Author Response

Dear Reviewers,

We would like to express our deepest gratitude for your invaluable assistance during the review process of our manuscript. Your thoughtful comments and insightful suggestions have significantly contributed to the improvement of our work.

Your thorough evaluation and constructive feedback have helped us refine the clarity and coherence of our research. We sincerely appreciate the time and expertise you have generously shared with us, as your input has been instrumental in strengthening the overall quality of our manuscript.

We have significantly improved the manuscript to reflect the new changes and suggestions provided as highlighted below: 

Reviewer 2:

The main aims of the studies performed by authors were to determine whether circularity, solidity, and fractal dimension of the cell clusters in liquid-based cytology are related to endometrial cancer (EC) or atypical endometrial hyperplasia (AEH), and can be used to enhance the accuracy of cytology diagnosis. 

The study's second aim was to determine whether this method can predict the prognosis of endometrial carcinoma. The paper is well written, the experiments performed are logical, and the results and conclussion are well presented

The results presented by the authors in their manuscript suggest that cell cluster analysis with liquid-endometrial cytology may improve diagnostic accuracy and prognosis prediction. 

Circularity and fractal analysis combined with endometrial cytology may contribute to detecting EC cases more accurately than endometrial cytology alone. Thus, circularity and fractal analysis may contribute to the early detection of EC and more than this, the fractal dimension has the potential to predict the prognosis for endometrial cancer.  All these represent the novelty of this article, and for these reasons, I recommend the publication of the manuscript.

One minor observation: a sentence cannot begin with an abbreviation, and in the manuscript, I detected more sentences in this situation: one on page 5 (a sentence which begins with  ''X-tile software...''), one on page 6 (a sentence which begins with: ''AUC values...'')  and one at page 21 ( sentence with begin with ''COx...'') I recommend authors to read with attention the manuscript and rewrite sentences which begin with an abbreviation.

Response:

Thank you for pointing this out. The following revisions have been made to the specified lines in the manuscript:

Page 5, Line 8

The X-tile software was used to determine cut-off values for categorizing high or low circularity, solidity, and fractal dimension [42].

Page 6, Line 26

The AUC values between 0.6 and 0.7, 0.71 and 0.8, and greater than 0.8 indicate weak predictive, satisfactory, and good predictive abilities, respectively [44].

Page 12, Line 22

The Cox hazards regression analysis showed that circularity and solidity were associated with OS in univariable analysis (Table 5).

Reviewer 3 Report

Comments and Suggestions for Authors

Dear Authors,

Please find my comments below:

1. In the whole paper (from the introduction to the discussion section), please be precise and add information about which kind of EC you described - endometroid or non-endometroid EC. 

2. Randomization graph is a good idea.

3. There are large differences in numbers between groups, which may affect the reliability and power of the results. Please explain in your answer and manuscript what steps were taken to minimise the risk of misinterpretation.

4. It would be beneficial to include a sample size analysis in your paper. This will provide a more comprehensive understanding of your research.

5. Ethical consideration should be the first part of the M&M section.

6. Keywords should come from the mesh database.

7. References should be prepared according to the Journal's recommendation.

8. a letter "p" should be written in italics, i.e p < 0.05

9. Some typo mistakes should be eliminated.

10. I wonder if multiomics strategies can be helpful in your area of interest described in this study. Please discuss it and consider to citing Int J Mol Sci. 2022 Jan 22;23(3):1237. doi: 10.3390/ijms23031237.  

Author Response

Dear Reviewer,

We would like to express our deepest gratitude for your invaluable assistance during the review process of our manuscript. Your thoughtful comments and insightful suggestions have significantly contributed to the improvement of our work.

Your thorough evaluation and constructive feedback have helped us refine the clarity and coherence of our research. We sincerely appreciate the time and expertise you have generously shared with us, as your input has been instrumental in strengthening the overall quality of our manuscript.

We have significantly improved the manuscript to reflect the new changes and suggestions provided as highlighted below:

Reviewer 3:

Dear Authors,

Please find my comments below:

1. In the whole paper (from the introduction to the discussion section), please be precise and add  information about which kind of EC you described - endometroid or non-endometroid EC. 

Response:

In this paper, endometrial cancer (EC)  includes endometrioid cancer, serous cancer, and clear cell cancer. These inadequately described figure legends and tables have been corrected.  

2. Randomization graph is a good idea.

Responces:

Thank you for your suggestions. Regarding questions 3 and 4, our study met the power requirements of ROC analysis. We have added graphs illustrating sensitivity, specificity, and thresholds for each cell cluster in the ROC analysis.

Page 6, Line 33

Graphs depicting the relationship between thresholds and sensitivity or specificity for cell cluster measurements are provided in Supplemental Figures S1 and S2.

3. There are large differences in numbers between groups, which may affect the reliability and power of the results. Please explain in your answer and manuscript what steps were taken to minimise the risk of misinterpretation.

Responces:

Previous studies have indicated that negative cytology cases are rarely malignant. In this study, negative cytology cases were randomly selected, resulting in an almost equal number of AEH+EC and Normal+EH cases for histology results. The AUC for the diagnostic accuracy of EC+AEH is unknown due to the lack of previous studies on cell clusters. Therefore, we conducted a preliminary analysis to predict the AUC for EC+AEH diagnostic accuracy and to determine if the sample size meets the following criteria: a significance level of 5%, 90% power, a two-tailed test, and a ratio of normal to abnormal cases of 1:1.

Page 3, Line 36

As a result, 91 endometrial cytology-suspicion-positive cases and 83 endometrial cytology-positive cases matching these criteria were included in the analysis. In this study, 26 negative cytology cases were randomly selected, resulting in an almost equal number of AEH+EC and Normal + EH cases for histology results.

Page 6, Line 27

The AUC for the diagnostic accuracy of EC+AEH is unknown due to the lack of previous studies on cell clusters. Therefore, we conducted a sample size analysis to determine if the sample size meets the following criteria: a significance level of 5%, 90% power, a two-tailed test, and a ratio of normal to abnormal cases of 1:1.

4. It would be beneficial to include a sample size analysis in your paper. This will provide a more comprehensive understanding of your research.

Responses:

In terms of fractal dimension, the preliminary AUC analysis for the diagnostic accuracy of EC+AEH is approximately 0.7. Assuming an AUC of 0.7 on the ROC curve, the required number of cases, calculated with a significance level of 0.5%, 90% power, a two-tailed test, and a ratio of normal to abnormal cases of 1:1, was 40. This study satisfactorily met these requirements.For circulality and solidity, the preliminary AUC analysis for the diagnostic accuracy of EC+AEH is approximately 0.65. Assuming an AUC of 0.65 on the ROC curve, the required number of cases, calculated with a significance level of 0.5%, 90% power, a two-tailed test, and a ratio of normal to abnormal cases of 1:1, was 73. This study satisfactorily met these requirements.

Page 6, Line 27

The AUC for the diagnostic accuracy of EC+AEH is unknown due to the lack of previous studies on cell clusters. Therefore, we conducted a sample size analysis to determine if the sample size meets the following criteria: a significance level of 5%, 90% power, a two-tailed test, and a ratio of normal to abnormal cases of 1:1. Assuming an AUC of 0.7 on the ROC curve, the required number of cases was 40. Assuming an AUC of 0.65 on the ROC curve, the required number of cases was 73. 

5. Ethical consideration should be the first part of the M&M section.

Response:

We have detailed the ethical considerations in the initial part of the Materials and Methods section.The following revisions have been made to the specified lines in the manuscript:

Page 3, Line 28

This study was conducted in adherence to the tenets of the Declaration of Helsinki.

6. Keywords should come from the mesh database.

Response:

We have listed the keywords come from the mesh database.The following revisions have been made to the specified lines in the manuscript:

Page 1, Line 53

Keywords: Endometrial Neoplasms; Cytodiagnosis; Cytological Techniques; Image Processing, Computer Assisted; Fractals; prognosis

7. References should be prepared according to the Journal's recommendation.

Response:

References have been prepared according to the Journal's recommendation. 

8. a letter "p" should be written in italics, i.e p < 0.05

Response:

The letter "p" has been italicized.

9. Some typo mistakes should be eliminated.

Response:

We have reviewed the entire paper to ensure there are no typographical errors.

10. I wonder if multiomics strategies can be helpful in your area of interest described in this study. Please discuss it and consider to citing Int J Mol Sci. 2022 Jan 22;23(3):1237. doi: 10.3390/ijms23031237. 

Response:

Thank you for the great suggestions. We believe that a multi-omics strategy is crucial for the development of novel therapies for endometrial cancer. By incorporating a morphological approach, as demonstrated in our study, will establish a foundation for innovative diagnostic and therapeutic methods. The following revisions have been made to the specified lines in the manuscript:

Page 15, Line 19

FIGO 2023 staging uses the molecular classification proposed by The Cancer Genome Atlas (TCGA), which classifies endometrial carcinomas into four categories based on the molecular classification: polymerase ε (POLE, ultramutated), microsatellite instability (MSI, hypermutated), copy number low, and copy number high [35,36]. The molecular classification is linked to the prognosis of endometrial cancer [35,36]. Multiomics approaches offer a comprehensive view at the molecular landscape of EC, aiding in the identification of markers and therapeutic targets. These methods integrate data from the genome, transcriptome, proteome, and epigenome, providing a broader understanding of the disease [60]. By incorporating a morphological approach, as demonstrated in our study, will establish a foundation for innovative diagnostic and therapeutic methods.

Round 2

Reviewer 3 Report

Comments and Suggestions for Authors

I have no further comments.